# FracTrain: Fractionally Squeezing Bit Savings Both Temporally and Spatially for Efficient DNN Training

**Yonggan Fu**[†], **Haoran You**[†] , **Yang Zhao**[†] , **Yue Wang**[†], **Chaojian Li**[†],
**Kailash Gopalakrishnan**[◇], **Zhangyang Wang**[‡], and **Yingyan Lin**[†]

[†]Department of Electrical and Computer Engineering, Rice University
[‡]Department of Electrical and Computer Engineering, The University of Texas at Austin
[◇]IBM T. J. Watson Research Center
[†]*{yf22, hy34, zy34, yw68, cl114, yingyan.lin}@rice.edu*
[‡]*atlaswang@utexas.edu*, [◇]*kailash@us.ibm.com*

## Abstract

Recent breakthroughs in deep neural networks (DNNs) have fueled a tremendous demand for intelligent edge devices featuring on-site learning, while the practical realization of such systems remains a challenge due to the limited resources available at the edge and the required massive training costs for state-of-the-art (SOTA) DNNs. As reducing precision is one of the most effective knobs for boosting training time/energy efficiency, there has been a growing interest in low-precision DNN training. In this paper, we explore from an orthogonal direction: how to fractionally squeeze out more training cost savings from the most redundant bit level, progressively along the training trajectory and dynamically per input. Specifically, we propose FracTrain that integrates *(i)* **progressive fractional quantization** which gradually increases the precision of activations, weights, and gradients that will not reach the precision of SOTA static quantized DNN training until the final training stage, and *(ii)* **dynamic fractional quantization** which assigns precisions to both the activations and gradients of each layer in an input-adaptive manner, for only "fractionally" updating layer parameters. Extensive simulations and ablation studies (six models, four datasets, and three training settings including standard, adaptation, and fine-tuning) validate the effectiveness of FracTrain in reducing computational cost and hardware-quantified energy/latency of DNN training while achieving a comparable or better (-0.12% ~ +1.87%) accuracy. For example, when training ResNet-74 on CIFAR-10, FracTrain achieves 77.6% and 53.5% computational cost and training latency savings, respectively, compared with the best SOTA baseline, while achieving a comparable (-0.07%) accuracy. Our codes are available at: https://github.com/RICE-EIC/FracTrain.

## 1 Introduction

Recent breakthroughs in deep neural networks (DNNs) have motivated an explosive demand for intelligent edge devices. Many of them, such as autonomous vehicles and healthcare wearables, require real-time and on-site learning to enable them to proactively learn from new data and adapt to dynamic environments. The challenge for such on-site learning is that the massive and growing cost of state-of-the-art (SOTA) DNNs stands at odds with the limited resources available at the edge devices, raising a major concern even when training in cloud using powerful GPUs/CPUs [1, 2].

To address the above challenge towards efficient DNN training, low-precision training have been developed recognizing that the training time/energy efficiency is a quadratic function of DNNs' adopted precision [3]. While they have showed promising training efficiency, they all adopt *(i)* a **static** quantization strategy, i.e., the precisions are fixed during the whole training process; *(ii)*

the **same** quantization precision for all training samples, limiting their achievable efficiency. In parallel, it has been recognized that different stages along DNNs' training trajectory require different optimization and hyperparameters, and not all inputs and layers are equally important/useful for training: [4] finds that DNNs which learn to fit different patterns at different training stages tend to have better generalization capabilities, supporting a common practice that trains DNNs starting with a large learning rate and annealing it when the model is fit to the training data plateaus; [5] reveals that some layers are critical to be intensively updated for improving the model accuracy, while others are insensitive, and [6, 7] show that different samples might activate different sub-models.

Inspired by the prior arts, we propose FracTrain, which **for the first time** advocates a progressive and dynamic quantization strategy during training. Specifically, we make the following contributions:

- We first propose **progressive fractional quantization** (PFQ) training in which the precision of activations, weights, and gradients increases gradually and will not reach the precision of SOTA static low-precision DNN training until the final training stage. We find that a lower precision for the early training stage together with a higher precision for the final stage consistently well balance the training space exploration and final accuracy, while leading to large computational and energy savings. Both heuristic and principled PFQ are effective.

- We then introduce **dynamic fractional quantization** (DFQ) training which automatically adapts precisions of different layers to the inputs. Its core idea is to hypothesize layer-wise quantization (to different precisions) as intermediate "soft" choices between fully utilizing and completely skipping a layer. DFQ's finer-grained dynamic capability consistently favors much better trade-offs between accuracy and training cost across different DNN models, datasets, and tasks, while being realized with gating functions that have a negligible overhead ($< 0.1\%$).

- We finally integrate PFQ and DFQ into one unified framework termed **FracTrain**, which is **the first** to adaptively squeeze out training cost from the finest bit level *temporally* and *spatially* during training. Extensive experiments show that FracTrain can aggressively boost DNN training efficiency while achieving a comparable or even better accuracy, over the most competitive SOTA baseline. Interestingly, FracTrain's effectiveness across various settings coincides with recent findings [4, 8, 5, 7, 9] that *(i)* different stages of DNN training call for different treatments and *(ii)* not all layers are equally important for training convergence.

## 2 Prior works

**Accelerating DNN training.** Prior works attempt to accelerate DNN training in resource-rich scenarios via communication-efficient distributed optimization and larger mini-batch sizes [10, 11, 12, 13]. For example, [11] combined distributed training with a mixed precision framework to train AlexNet in 4 minutes. While distributed training can reduce training time, it increases the energy cost. In contrast, we target energy efficient training for achieving in-situ, resource-constrained training.

**Low-precision training.** Pioneering works have shown that DNNs can be trained under low precision [14, 3, 15, 16, 17], instead of full-precision. First, distributed efficient learning reduces the communication cost of aggregation operations using quantized gradients [18, 19, 20, 21], which however cannot reduce training costs as they mostly first compute full-precision gradients and then quantize them. Second, low-precision training achieves a better trade-off between accuracy and efficiency towards on-device learning. For example, [3, 17] and [14, 22] introduced an 8-bit floating-point data format and a 8-bit integer representation to reduce training cost, respectively. FracTrain explores from an orthogonal perspective, and can be applied on top of them to further boost the training efficiency.

**Dynamic/efficient DNN training.** More recently *dynamic inference* [23, 9, 24, 25, 26, 27, 28, 29] was developed to reduce the average inference cost, which was then extended to the most fine-grained bit level [30, 31]. While energy-efficient training is more complicated than and different from inference, many insights of the latter can be lent to the former. For example, [32] recently accelerated the empirical convergence via active channel pruning during training; [33] integrated stochastic data dropping, selective layer updating, and predictive low-precision to reduce over 80% training cost; and [34] accelerated training by skipping samples that leads to low loss values per iteration. Inspired by these works, our proposed FracTrain pushes a new dimension of dynamic training via temporally and spatially skipping unnecessary bits during the training process.

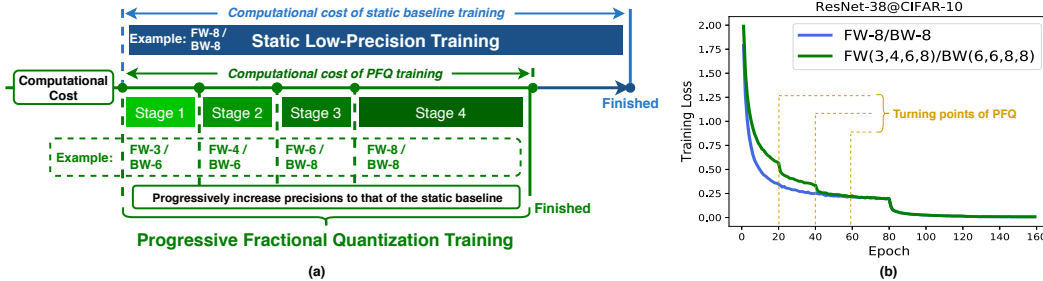

Figure 1: **(a)** A high-level view of the proposed PFQ vs. SOTA low-precision training, where PFQ adopts a four-stage precision schedule to gradually increase the precision of weights, activations, and gradients up to that of the static baseline which here employs 8-bit for both the forward and backward paths, denoted as FW-8/BW-8, and **(b)** the corresponding training loss trajectory.

## 3   The proposed techniques

This section describes our proposed efficient DNN training techniques, including PFQ (Section 3.1), DFQ (Section 3.2), and FracTrain that unifies PFQ and DFQ (Section 3.3).

### 3.1   Progressive Fractional Quantization (PFQ)

**Hypothesis.** The proposed PFQ draws inspiration from *(i)* [35, 36], which argue that DNNs first learn low-complexity (lower-frequency) functional components before absorbing high-frequency features, with the former being more robust to perturbations, and *(ii)* [8], which shows that training DNNs starting with a large initial learning rate helps to learn more generalizable patterns faster and better, i.e., faster convergence and higher accuracy. We hypothesize that precision of DNNs can achieve similar effects, i.e., a lower precision in the early training stage fits the observed behavior of learning lower-complexity, coarse-grained patterns, while increasing precision along with training trajectory gradually captures higher-complexity, fine-grained patterns. In other words, staying at lower precisions implies larger quantization noise at the beginning, that can inject more perturbations to favor more robust exploration of the optimization landscape. Therefore, it is expected that DFQ can boost training efficiency while not hurting, or even helping, the model's generalization performance.

**Design of PFQ.** We propose PFQ that realizes the aforementioned hypothesis in a principled manner by developing a simple yet effective indicator to automate PFQ's precision schedule, as described in *Algorithm* 1. Specifically, we measure the difference of the normalized loss function in consecutive epochs, and increase the precisions when the loss difference in the previous five epochs is smaller than a preset threshold $\epsilon$; We also scale $\epsilon$ by a decaying factor $\alpha$ to better identify turning points, as the loss curve proceeds to the final plateau. The proposed indicator adapts $\epsilon$ proportionally w.r.t. the loss function's peak magnitude, and thus can generalize to different datasets, models, and tasks. In addition, it has negligible overhead (< 0.01% of the total training cost). Note that PFQ schedules precision during training based on prior works' insights on DNN training: *(i)* Gradients often require a higher precision than weights and activations [37], and *(ii)* more precise update (i.e., a higher precision) at the end of the training process is necessary for better convergence [38].

Fig.1 (a) shows a high-level view of PFQ as compared to static low-precision training, with Fig.1 (b) plotting an example of the corresponding training loss trajectory. In the example of Fig.1, we adopt a four-stage precision schedule for the early training phase (here referring to the phase before the first learning rate annealing at the 80-th epoch): the first stage assigns 3-bit and 6-bit precisions for the forward (i.e., weights and activations) and backward (i.e., gradients) paths, denoted as FW-3/BW-6; The final stage employs a precision of FW-8/BW-8, which is the same as that of the static low-precision training baseline; and the intermediate stages are assigned precision that uniformly interpolates between that of the first and final stages. PFQ in this example achieves 63.19% computational cost savings over the static training baseline, while improving the accuracy by 0.08%. Note that we assume integer-based quantization format [14] in this example.

**Insights.** PFQ reveals a new "fractional" quantization progressively along the training trajectory: SOTA low-precision DNN training that quantizes or skips the whole model can be viewed as the two "extremes" of quantization (i.e., full vs. zero bits), while training with the intermediate precision attempts to "fractionally" quantize/train the model. As shown in Fig.1 and validated in Section 4.2, PFQ can automatically squeeze out unnecessary bits from the early training stages to simultaneously boost training efficiency and accuracy, while being simple enough for easy adoption.

| **Algorithm 1:** Progressive Fractional Quantization Training | **Algorithm 2:** FracTrain: Integrating PFQ and DFQ |
|---|---|
| 1: Initialize the precision schedule $\{bit_i\}$ (initial $i = 0$), indicating threshold $\epsilon$, decaying factor $\alpha$, training epoch $max\_epoch$ | 1: Initialize target $cp$ schedule $\{cp_i\}$ (initial $i = 0$), indicating threshold $\epsilon$, decaying factor $\alpha$, training epoch $max\_epoch$ |
| 2: **while** epoch $< max\_epoch$ **do** | 2: **while** epoch $< max\_epoch$ **do** |
| 3:    Training with precision setting $bit_i$ for one epoch | 3:    **for** training one epoch **do** |
| 4:    Calculate normalized loss difference $Loss\_diff$ between consecutive epochs | 4:      Optimize Eq. (2) with DFQ |
| 5:    **if** $Loss\_diff < \epsilon$ **then** | 5:      Adaptively flip the sign of $\beta$ in Eq. (2) |
| 6:      $i \leftarrow i + 1$ (switch to next $bit_{i+1}$) | 6:    **end for** |
| 7:      decay $\epsilon$ by $\alpha$ (prepare for next switch) | 7:    Get $Loss\_diff$ as in *Algorithm* 1 |
| 8:    **end if** | 8:    **if** $Loss\_diff < \epsilon$ **then** |
| 9: **end while** | 9:      $i \leftarrow i + 1$ (switch to next $cp_{i+1}$) |
| | 10:      decay $\epsilon$ by $\alpha$ (prepare for next switch) |
| | 11:    **end if** |
| | 12: **end while** |

## 3.2 Dynamic Fractional Quantization (DFQ)

**Hypothesis.** We propose DFQ to dynamically adapt precisions of activations and gradients in an input-dependent manner. Note that SOTA DNN hardware accelerators have shown that dynamic precision schemes are **hardware friendly**. For example, [39] developed a bit-flexible DNN accelerator that constitutes bit-level processing units to dynamically match the precision of each layer. With such dedicated accelerators, DFQ's training cost savings would be maximized.

**Design of DFQ.** To our best knowledge, DFQ is the first attempt to unify the binary selective layer update design and quantization into one unified training framework in order to dynamically construct intermediate "soft" variants of selective layer update. Fig. 2 illustrates our DFQ framework, in which the operation of a DNN layer can be formulated as:

$$F_i = \sum_{n=1}^{N-1} G_i^n C_i^{b_n}(F_{i-1}) + G_i^0 F_{i-1} \quad (1)$$

where we denote *(i)* the output and input of the $i$-th layer as $F_i$ and $F_{i-1}$, respectively, *(ii)* the

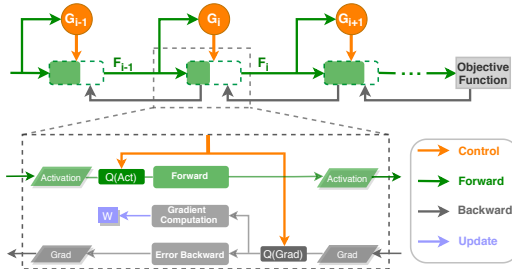

Figure 2: Illustrating the proposed DFQ on top of the $i$-th DNN layer/block, where the orange circle $G$ indicates a recurrent neural network (RNN) gate and Q $(\cdot)$ denotes a quantization operation.

convolution operation of the $i$-th layer executed with $k$ bits as $C_i^k$, where a gating network $G_i$ determines the fractional quantization precision, *(iii)* $G_i^n \in \{0, 1\}$ as the $n$-th entry of $G_i$, and *(iv)* $b_n$ as the precision option of the $n$-th entry (e.g., $n = 0$ or $N-1$ represents precisions of zero or full bits). Note that only one of the precision in $b_n (n = 0, .., N-1)$ will be activated during each iteration.

For designing the gating network, we follow [33] to incorporate a light-weight RNN per layer/block, which takes the same input as its corresponding layer, and outputs soft-gating probabilistic indicators. The highest-probability precision option is selected to train at each iteration. The RNN gates have a negligible overhead, e.g., <0.1% computational cost of the base layer/block.

**Training of DFQ.** To train DFQ, we incorporate a cost regularization into the training objective:

$$\min_{(W_{base}, W_G)} L(W_{base}, W_G) + \beta \times cp(W_{base}, W_G) \quad (2)$$

where $L$, $cp$, and $\beta$ denote the accuracy loss, the cost-aware loss, and the weighting coefficient that trades off the accuracy and training cost, respectively, and $W_{base}$ and $W_G$ denote weights of the backbone and gating networks, respectively. The cost-aware loss $cp$ in this paper is defined as the ratio of the computational cost between the quantized and full-precision models in each training iteration. To achieve a specified $cp$, DFQ automatically controls the sign of $\beta$: if $cp$ is higher than the specified one, $\beta$ is set to be positive, enforcing the model to reduce its training cost by suppressing $cp$ in Eq. (2); if $cp$ is below the specified one, the sign of $\beta$ is flipped to be negative, encouraging the model to increase its training cost. In the end, $cp$ will stabilize around the specified value.

**Insights.** DFQ unifies two efficient DNN training mindsets, i.e., dynamic selective layer update and static low-precision training, and enables a "fractional" quantization of layers during training, in contrast to either a full execution (selected) or complete non-execution (bypassed) of layers. Furthermore, DFQ introduces *input-adaptive quantization* at training for the first time, and automatically learns to adapt the precision of different layers' activations and gradients in contrast to current practice of low-precision training [14, 22, 37] that fixes layer-wise precision during training regardless of inputs. In effect, the selective layer update in [33] can be viewed as a coarse-grained version of DFQ, i.e., allowing only to select between full bits (executing without quantization) and zero bits (bypassed).

### 3.3 FracTrain: unifying PFQ and DFQ

PFQ and DFQ explore two orthogonal dimensions for adaptive quantization towards efficient training: "*temporally*" along the training trajectory, and "*spatiallly*" for the model layout. It is hence natural to integrate them into one unified framework termed FracTrain, that aims to maximally squeeze out unnecessary computational cost at the finest granularity. The integration of PFQ and DFQ in FracTrain is straightforward and can be simply realized by applying PFQ to DFQ based models: the DFQ based training process is automatically divided into multiple stages controlled by the PFQ indicator in *Algorithm* 1 and each stage is assigned with different target $cp$, thus squeezing more bit savings from the early training stages. *Algorithm* 2 summarizes the design of our FracTrain.

## 4 Experiments

We will first present our experiment setup in Section 4.1 and the ablation studies of FracTrain (i.e., evaluating PFQ and DFQ) in Sections 4.2 and 4.3 respectively, and then FracTrain evaluation under different training settings in Section 4.4 and 4.5. Finally, we will discuss the connections of FracTrain with recent theoretical analysis of DNN training and the ML accelerators to support FracTrain.

### 4.1 Experiment setup

**Models, datasets, and baselines.** Models & Datasets: We consider a total of **six DNN models** (i.e., ResNet-18/34/38/74 [40], MobileNetV2 [41], and Transformer-base [42]) and **four datasets** (i.e., CIFAR-10/100 [43], ImageNet [44], and WikiText-103 [45]). Baselines: We evaluate FracTrain against three SOTA static low-precision training techniques, including SBM [14], WAGEUBN [22], and DoReFa [37], and perform ablation studies of FracTrain (i.e., evaluation of PFQ and DFQ) over SBM [14] which is the most competitive baseline based on both their reported and our experiment results. Note that we keep all the batch normalization layers [46] in floating point precision in all experiments for our techniques and the baselines, which is a common convention in literature.

**Training settings.** For training, we follow SOTA settings in [9] for experiments on CIFAR-10/100 and [9] for experiments on ImageNet, for which more details are provided in the supplement. For the hyperparameters of FracTrain, we simply calculate $\epsilon$ and $\alpha$ (0.05 and 0.3, respectively) from the normalized loss around the turning points for a four-stage PFQ on ResNet-38 with CIFAR-10 (see Fig. 1), and then apply them to all experiments. The resulting $\epsilon$ and $\alpha$ work for all the experiments, showing FracTrain's insensitivity to its hyper hyperparameters.

**Evaluation metrics.** We evaluate PFQ, DFQ, and FracTrain in terms of the following cost-aware metrics of training costs, in addition to the model *accuracy (Acc)*: *(i) Computational Cost (CC):* Inspired by [37] and following [30], we calculate the computational cost of DNNs using the effective number of MACs, i.e., (# of $MACs) * Bit_a/32 * Bit_b/32$ for a dot product between $a$ and $b$, where $Bit_a$ and $Bit_b$ denote the precision of $a$ and $b$, respectively. As such, this metric is proportional to the total number of bit operations. *(ii) Energy and Latency:* The *CC* might not align well with the actual energy/latency in real hardware [47], we thus also consider training energy and latency characterized using a SOTA cycle-accurate simulator, named BitFusion, based on Register-Transfer-Level (RTL) implementations in a commercial CMOS technology and a SOTA DNN accelerator that supports arbitrary precisions [48]. Since backpropagation can be viewed as two convolution processes (for computing the gradients of weights and activations, respectively), we estimate the training energy and latency by executing the three convolution processes sequentially in BitFusion. Note that we apply BitFusion for both our FracTrain and all the integer-only baselines to make sure that the adopted hardware parameters (e.g., dataflows) are the same for a fair comparison.

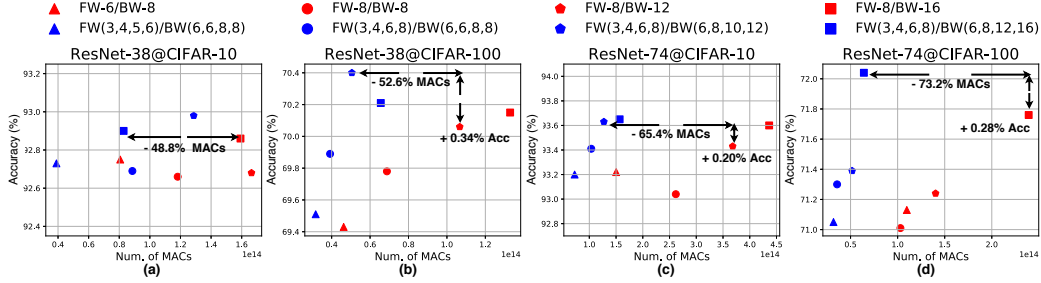

Figure 3: Comparing PFQ (blue) with the most competitive baseline, SBM [14] (red), in terms of model accuracy vs. the total number of training MACs on ResNet-38/74 models with CIFAR-10/100. Note that we use FW-6/BW-8 to denote FW(6,6,6,6)/BW(8,8,8,8) for short.

## 4.2 FracTrain ablation study: evaluate PFQ

This subsection evaluates PFQ over the most competitive baseline, SBM [14].

**PFQ on ResNet-38/74 and CIFAR-10/100.** Fig. 3 compares the accuracy vs. the total number of MACs of SBM and PFQ on ResNet-38/74 and CIFAR-10/100 under four different precision schemes. We have **two observations**. First, PFQ consistently outperforms SBM [14] by reducing the training cost while achieving a comparable or even better accuracy. Specifically, PFQ reduces the training cost by 22.7% ~ 73.2% while offering a comparable or better accuracy (-0.08% ~ +0.34%), compared to SBM. For example, when training ResNet-74 on CIFAR-100, PFQ of FW(3,4,6,8)/BW(6,8,12,16) achieves 73.2% computational savings and a better (+0.28%) accuracy over SBM of FW(8,8,8,8)/BW(16,16,16,16). Second, PFQ achieves larger computational cost savings when the models target a higher accuracy and thus require a higher precision.

Experiments under more precision settings are provided in the supplement.

**Sensitivity to hyperparameters in PFQ.** To verify the sensitivity of PFQ to its hyperparameters, we evaluate PFQ of FW(3,4,6,8)/BW(6,6,8,8) for training ResNet-38 on CIFAR-100 as shown in Fig. 4 under various $\epsilon$ (different shapes) and $\alpha$ (different colors). We can see that a good accuracy-efficiency trade-off can be achieved by PFQ in a large range of hyperparameter settings as compared with its static baselines, showing PFQ's insensitivity to hyperparameters. It is intuitive that (1) $\epsilon$ and $\alpha$ control the accuracy-efficiency trade-off, and (2) a larger $\epsilon$ and $\alpha$ (i.e., faster precision increase) lead to a higher training cost and higher accuracy. We also show PFQ's insensitivity to its precision schedule under three different precision schedule strategies in the supplement.

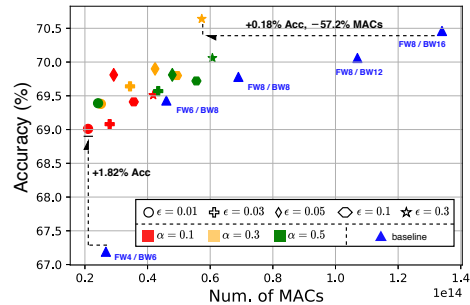

Figure 4: PFQ of FW(3,4,6,8)/BW(6,6,8,8) vs. SBM on ResNet-38 and CIFAR-100 under various $\epsilon$ (see shapes) and $\alpha$ (see colors).

**PFQ on MobileNetV2 and CIFAR-10/100.** We also evaluate PFQ on compact DNN models. Table 1 shows that PFQ's benefits even extend to training compact models such as MobileNetV2. Specifically, as compared to SBM under three precision schedule schemes, PFQ achieves computational cost savings of 27.4% ~ 49.8%

Table 1: PFQ vs. SBM on MobileNetV2 and CIFAR-10/100.

| Precision Setting | DataSet | Acc ($\Delta$Acc) | MACs | Comp. Saving |
|---|---|---|---|---|
| FW(4,4,6,8)/ | CIFAR-10 | 93.77 (+0.04%) | 1.22E+14 | 17.16% |
| BW(6,6,8,8) | CIFAR-100 | 74.84 (+0.09%) | 6.33E+13 | 27.41% |
| FW(4,4,6,8)/ | CIFAR-10 | 93.69 (+0.03%) | 1.40E+14 | 27.30% |
| BW(6,8,10,12) | CIFAR-100 | 75.07 (+0.37%) | 6.69E+13 | 44.94% |
| FW(4,4,6,8)/ | CIFAR-10 | 93.90 (-0.10%) | 1.87E+14 | 21.96% |
| BW(6,8,12,16) | CIFAR-100 | 74.94 (+0.03%) | 7.60E+13 | 49.84% |

and 17.2% ~ 27.3%, while having a comparable or better accuracy of -0.10% ~ +0.04% and -0.10% ~ +0.04%, respectively, on the CIFAR-10 and CIFAR-100 datasets. For experiments using MobileNetV2 in this paper, we adopt a fixed precision of FW-8/BW-16 for depthwise convolution layers as MobileNetV2's accuracy is sensitive to the precision of its separable convolution.

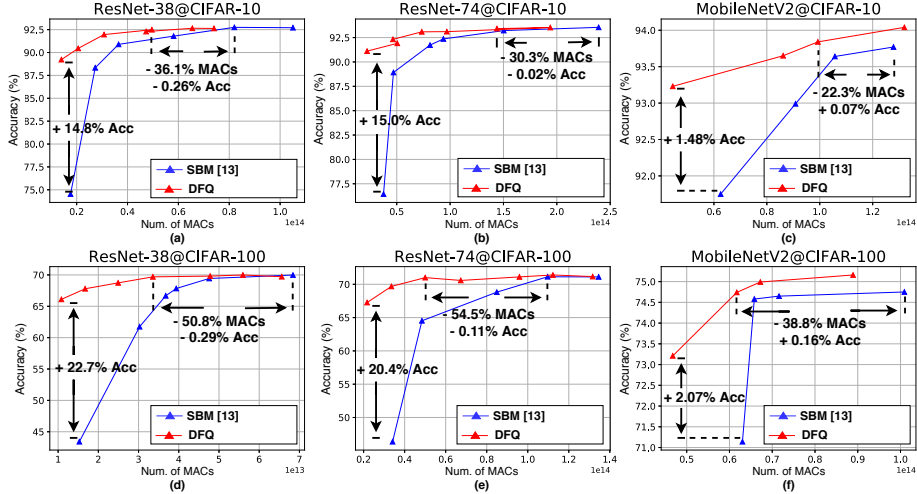

Figure 5: Comparing DFQ with SBM [14] on ResNet-38/74/MobileNetV2 and CIFAR-10/100.

**PFQ on ImageNet and WikiText-103.** We then evaluate PFQ on *(i)* a large vision dataset ImageNet and *(ii)* a language modeling dataset WikiText-103 to verify its general effectiveness. Table 2 shows that PFQ again outperforms SBM on both tasks: PFQ reduces the computational cost by 21.44% on the relatively small

Table 2: PFQ vs. SBM on ResNet-18/ImageNet and Transformer-base/WikiText-103.

| Model / Dataset | Method | Precision Setting | Acc (%) / Perplexity | MACs |
|---|---|---|---|---|
| ResNet-18 ImageNet | SBM | FW-8 / BW-16 | 69.51 | 3.37E+15 |
| | PFQ | FW(3,4,6,8) / BW(6,8,12,16) | 69.68 | 2.64E+15 |
| | | **PFQ Improv.** | **+0.17** | **21.44%** |
| Transformer-base WikiText-103 | SBM | FW-8 / BW-16 | 31.55 | 2.81E+14 |
| | PFQ | FW(3,4,6,8) / BW(6,8,12,16) | 31.49 | 1.57E+14 |
| | | **PFQ Improv.** | **-0.06** | **44.0%** |

ResNet-18 while improving the accuracy by 0.17% on ImageNet and decreases the computational cost by 44.0% on Transformer-base/WikiText-103 while improving the perplexity by 0.06, as compared to the competitive SBM baseline.

Notably, **PFQ even achieves a higher accuracy than the SOTA floating-point training technique** [40] under most of the aforementioned experiments including ResNet-38/74 on CIFAR-10/100 and ResNet-18 on ImageNet, demonstrating PFQ's excellent generalization performance.

### 4.3 FracTrain ablation study: evaluate DFQ

This subsection evaluates the proposed DFQ over SBM [14] on three DNN models (RestNet-38/74 and MobileNetV2) and two datasets (CIFAR-10 and CIFAR-100), as shown in Fig. 5. We can see that DFQ surpasses SBM from two aspects: First, DFQ always demands less computational cost (i.e., the total number of MACs) to achieve the same or even better accuracy, on both larger models ResNet-38/74 and the compact model MobileNetV2; Second, while the static training baselines' performance deteriorates under very low computational costs, DFQ maintains decent accuracies, indicating that DFQ can achieve a better allocation of precision during training. Specifically, DFQ reduces the computational cost by 54.5% under a comparable accuracy (-0.11%), or boosts the accuracy by 22.7% while reducing the computational cost by 28.5%. Note that DFQ significantly outperforms the selective layer update in [33], e.g., achieving 7.3× computational savings with a higher accuracy on ResNet-38 and CIFAR-10, validating our hypothesis that DFQ's intermediate "soft" variants of selective layer update favor better trade-offs between accuracy and training costs.

More details for the experiment settings of Fig. 5 are provided in the supplement.

### 4.4 FracTrain over SOTA low-precision training

We next evaluate FracTrain over three SOTA low-precision training baselines including SBM [14], DoReFa [37], and WAGEUBN [22]. Here we consider standard training settings. FracTrain's bit allocation visualization are provided in the supplement.

Table 3: The training accuracy, computational cost, energy, and latency of FracTrain, SBM [14], WAGEUBN [22], and DoReFa [37], when training the ResNet-38/74 models on CIFAR-10/100.

| Model / Dataset | Method | Precision Setting | Acc (%) | MACs | Energy (kJ) | Latency (min) |
|---|---|---|---|---|---|---|
| ResNet-38 CIFAR-10 | WAGEUBN | FW-8 / BW-8 | 91.81 | 1.11E+14 | 31.63 | 292.74 |
| | DoReFa | FW-8 / BW-8 | 92.02 | 1.21E+14 | 34.34 | 317.83 |
| | SBM | FW-8 / BW-8 | **92.66** | 1.18E+14 | 33.55 | 310.47 |
| | DFQ | $cp$=3 | 92.49 | 4.96E+13 | 23.97 | 230.32 |
| | FracTrain | $cp$-1.5/2/2.5/3 | 92.54 | **3.97E+13** | **22.93** | **221.13** |
| | **FracTrain Improv.** | | -0.12 | **64.4 ~ 67.2%** | **27.5 ~ 33.2%** | **24.5 ~ 30.4%** |
| ResNet-38 CIFAR-100 | WAGEUBN | FW-8 / BW-8 | 67.95 | 1.15E+14 | 32.76 | 303.19 |
| | DoReFa | FW-8 / BW-8 | 68.63 | 1.01E+14 | 25.31 | 234.19 |
| | SBM | FW-8 / BW-8 | 69.78 | 6.88E+13 | 19.55 | 180.93 |
| | DFQ | $cp$=3 | 69.81 | 3.23E+13 | 16.22 | 155.91 |
| | FracTrain | $cp$-1.5/2/2.5/3 | **69.82** | **2.66E+13** | **15.74** | **151.87** |
| | **FracTrain Improv.** | | +0.04 | **61.3 ~ 77.0%** | **19.6 ~ 51.9%** | **16.1 ~ 49.9%** |
| ResNet-74 CIFAR-10 | WAGEUBN | FW-8 / BW-8 | 91.35 | 2.38E+14 | 68.21 | 629.58 |
| | DoReFa | FW-8 / BW-8 | 91.16 | 2.33E+14 | 66.84 | 616.90 |
| | SBM | FW-8 / BW-8 | 93.04 | 2.62E+14 | 75.01 | 692.29 |
| | DFQ | $cp$=3 | **93.09** | 7.33E+13 | 35.88 | 343.76 |
| | FracTrain | $cp$-1.5/2/2.5/3 | 92.97 | **5.85E+13** | **33.40** | **321.98** |
| | **FracTrain Improv.** | | +0.05 | **74.9 ~ 77.6%** | **50.0 ~ 55.5%** | **47.8 ~ 53.5%** |
| ResNet-74 CIFAR-100 | WAGEUBN | FW-8 / BW-8 | 69.61 | 1.34E+14 | 38.46 | 354.93 |
| | DoReFa | FW-8 / BW-8 | 69.31 | 1.79E+14 | 51.28 | 473.24 |
| | SBM | FW-8 / BW-8 | 71.01 | 1.40E+14 | 40.08 | 369.94 |
| | DFQ | $cp$=3 | 70.58 | 6.72E+13 | 32.89 | 315.11 |
| | FracTrain | $cp$-1.5/2/2.5/3 | **71.03** | **5.46E+13** | **32.23** | **310.67** |
| | **FracTrain Improv.** | | +0.02 | **59.3 ~ 69.5%** | **16.2 ~ 37.1%** | **12.5 ~ 34.4%** |

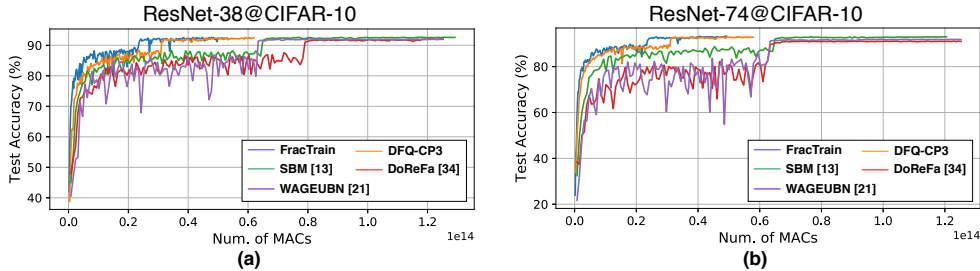

Figure 6: The testing accuracy's evolution along the training trajectories of different low-precision training schemes on ResNet-38/74 with CIFAR-10, where the x-axis captures the total computational cost up to each epoch.

**Accuracy and training costs.** Table 3 compares FracTrain in terms of accuracy and various training costs (computational cost, energy, and latency) against the three baselines when training ResNet-38/74 with CIFAR-10/100, where we use *FracTrain Improv.* to record the performance improvement of FracTrain over **the strongest competitor** among the three SOTA baselines. We can see that FracTrain **consistently outperforms** all competitors by reducing training computational cost, energy, and latency, while improving the accuracy in most cases, under all the models and datasets. Specifically, FracTrain can achieve training cost savings of 59.3 ~ 77.6%, 16.2% ~ 55.5%, and 12.5% ~ 53.5% in terms of the computational cost, energy, and latency, while leading to a comparable or even better accuracy (-0.12% ~ +1.87%). Furthermore, we also evaluate FracTrain using ResNet-18/34 on ImageNet (see Table 4). Again, we can see that FracTrain reduces the training computational cost (by 52.85% and 51.90% respectively) while achieving a comparable accuracy.

Table 4: FracTrain vs. SBM on ImageNet.

| Model | Method | Precision Setting | Acc (%) | MACs |
|---|---|---|---|---|
| ResNet-18 | SBM | FW-8 / BW-16 | 69.51 | 3.37E+15 |
| | FracTrain | $cp$=3/3.5/4/4.5 | 69.44 | 1.59E+15 |
| | **PFQ Improv.** | | **-0.07** | **52.85%** |
| ResNet-34 | SBM | FW-8 / BW-16 | 73.34 | 7.18E+15 |
| | FracTrain | $cp$=3/3.5/4/4.5 | 73.03 | 3.45E+15 |
| | **PFQ Improv.** | | **-0.31** | **51.90%** |

**Training trajectory.** Fig. 6 visualizes the testing accuracy's trajectories of FracTrain ($cp$=1.5/2/2.5/3), DFQ ($cp$=3), and the three baselines as the training computational cost increases on the ResNet-38/74 models and CIFAR-10 dataset, where DFQ-CP3 denotes the DFQ training with $cp$ =3%. We can see that FracTrain reaches the specified accuracy given the least training computational cost.

Table 5: Adaptation & fine-tuning training performance of the proposed PFQ, DFQ, FracTrain, and the SBM baseline [14] when training ResNet-38 on CIFAR-100's subsets.

| Model / Dataset | Method | Precision Setting | Adaptation | | Fine-tuning | |
|---|---|---|---|---|---|---|
| | | | Acc (%) | MACs | Acc (%) | MACs |
| | SBM | FW-8 / BW-8 | 77.44 | 9.96E+13 | 64.83 | 1.22E+14 |
| | PFQ | FW(3,4,6,8) / BW(6,6,8,8) | 77.76 | 7.87E+13 | 64.72 | 8.68E+13 |
| ResNet-38 | DFQ | $cp$=3 | **78.08** | 4.96E+13 | **65.01** | **3.95E+13** |
| CIFAR-100 | FracTrain | $cp$-1.5/2/2.5/3 | 77.52 | **3.12E+13** | 64.53 | 4.45E+13 |
| | **FracTrain Improv.** | | **+0.08%** | **68.7%** | **-0.3%** | **67.6%** |

## 4.5 FracTrain on adaptation & fine-tuning scenarios

To evaluate the potential capability of FracTrain for on-device learning [49], we consider training settings of both adaptation and fine-tuning, where the detailed settings are described in the supplement.

Table 5 compares the proposed PFQ, DFQ, and FracTrain with the SBM baseline in terms of accuracy and the computational cost in the adaptation  fine-tuning stage, i.e., the highest accuracy achieved during retraining and the corresponding computational cost. We can see that PFQ, DFQ, and FracTrain can all achieve a better or comparable accuracy over SBM, while leading to a large computational cost savings. Specifically, FracTrain reduces the training cost by 68.7% and 67.6% while offering a better (+0.08%) or comparable (-0.3%) accuracy, as compared to the SBM baseline for adaptation and fine-tuning training, respectively.

## 4.6 Discussions

**Connections with recent theoretical findings.** There have been growing interests in understanding and optimizing DNN training. For example, [35, 36] advocate that DNN training first learns low-complexity (lower-frequency) functional components and then high-frequency features, with the former being less sensitive to perturbations; [4] argues that important connections and the connectivity patterns between layers are first discovered at the early stage of DNN training, and then becomes relatively fixed in the latter training stage, which seems to indicate that critical connections can be learned independent of and also ahead of the final converged weights; and [8] shows that training DNNs with a large initial learning rate helps the model to memorize easier-to-fit and more generalizable pattern faster and better. Those findings regarding DNN training seem to be consistent with the effectiveness of our proposed FracTrain.

**ML accelerators to support FracTrain.** Both dedicated ASIC [50, 51] or FPGA accelerators (e.g., EDD [52]) can leverage the required lower average precision of FracTrain to reduce both the data movement and computation costs during training. As an illustrative example, we implement FracTrain on FPGA to evaluate its real-hardware benefits, following the design in EDD [52], which adopts a recursive architecture for mixed precision networks (i.e., the same computation unit is reused by different precisions) and a dynamic logic to perform dynamic schedule. The evaluation results using ResNet-38/ResNet-74 on CIFAR-100 and evaluated on a SOTA FPGA board (Xilinx ZC706 [53]) show that FracTrain leads to 34.9%/36.6% savings in latency and 30.3%/24.9% savings in energy as compared with FW8-/BW-8, while achieving a slightly better accuracy as shown in Table 3.

## 5 Conclusion

We propose a framework called FracTrain for efficient DNN training, targeting at squeezing out computational savings from the most redundant bit level along the training trajectory and per input. We integrate two dynamic low-precision training methods in FracTrain, including Progressive Fractional Quantization and Dynamic Fractional Quantization. The former gradually increases the precision of weights, activations, and gradients during training until reaching the final training stage; The latter automatically adapts precisions of different layers' activations and gradients in an input-dependent manner. Extensive experiments and ablation studies verify that our methods can notably reduce computational cost during training while achieving a comparable or even better accuracy. Our future work will strive to identify more theoretical grounds for such adaptively quantized training.

## Broader impact

Our FracTrain framework will potentially have a deep social impact due to its impressive efforts on efficient DNN training, which will greatly contribute to the popularization of Artificial Intelligence in daily life.

Efficient DNN training techniques are necessary from two aspects. For one thing, recent breakthroughs in deep neural networks (DNNs) have motivated an explosive demand for intelligent edge devices. Many of them, such as autonomous vehicles and healthcare wearables, require real-time and on-site learning to enable them to proactively learn from new data and adapt to dynamic environments. The challenge for such on-site learning is that the massive and growing cost of state-of-the-art (SOTA) DNNs stands at odds with the limited resources available at the edge devices. With the development of efficient training techniques, on-site learning becomes more efficient and economical, enabling pervasive intelligent computing systems like smart phones or smart watches in our daily life which deeply influences the life style of the whole society.

From another, despite the substantially growing need of on-device learning, current practices mostly train DNN models in a cloud server, and then deploy the pre-trained models into the devices for inference, due to the large gap between the devices' constrained resource and the highly complex training process. However, based on a recent survey training a DNN will generate five cars' life time carbon dioxide emission which is extremely environmental unfriendly. Efficient training techniques will notably help mitigate the negative ecological influence when training DNNs at data centers during the evolution of the AI field, which further boosts the high-speed development of AI and deepens its influences on the society.

Therefore, as the proposed FracTrain framework has been verified to be effective on various applications, its contribution to the efficient training field will directly bring positive impacts to the society. However, due to more pervasive applications driven by AI enabled by efficient training techniques, personal data privacy can be a potential problem which needs the help of other privacy-protecting techniques or regulations.

## Acknowledgement

We would like to thank Mr. Xiaofan Zhang at UIUC for his useful discussions and suggestions in our evaluation of FracTrain's training savings on FPGA when implemented using their EDD design. The work is supported by the National Science Foundation (NSF) through the Real-Time Machine Learning (RTML) program (Award number: 1937592, 1937588).

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
