[Supplementary Material]

# Supplementary Materials
# FracTrain: Fractionally Squeezing Bit Savings Both Temporally and Spatially for Efficient DNN Training

Yonggan Fu[†], Haoran You[†], Yang Zhao[†], Yue Wang[†], Chaojian Li[†],
Kailash Gopalakrishnan[◇], Zhangyang Wang[‡], and Yingyan Lin[†]

[†]Department of Electrical and Computer Engineering, Rice University
[‡]Department of Electrical and Computer Engineering, The University of Texas at Austin
[◇]IBM T. J. Watson Research Center
[†]{yf22, hy34, zy34, yw68, cl114, yingyan.lin}@rice.edu
[‡]atlaswang@utexas.edu, [◇]kailash@us.ibm.com

## 1 Ablation study about different PFQ strategies

Figure 1: Comparing PFQ with the most competitive baseline, SBM [1] (red), in terms of model accuracy vs. the total number of training MACs on ResNet-38/74 with CIFAR-10/100, where three variants of PFQ under four precision settings are considered. Note that we use FW-6/BW-8 to denote FW(6,6,6,6)/BW(8,8,8,8) for short.

To evaluate the general effectiveness of the proposed PFQ, here we compare three different PFQ strategies, including one heuristic PFQ (termed as manual-PFQ) and two principled PFQ (termed as Auto-PFQ). In particular, for the heuristic PFQ, we uniformly split the training process into four stages, and the principled PFQ (i.e., two-stage/four-stage Auto-PFQ) differs in the number of stages with progressive precisions as controlled by the loss indicator (see Section 3.1).

From the results shown in Fig. 1, we can observe that: First, both the heuristic and principled PFQ outperform SBM [1] by reducing the training cost while achieving a comparable or even better accuracy. Specifically, the Manual-PFQ with four stages and the Auto-PFQ with two and four stages achieve a reduced training cost of 14.8% ~ 64.0%, 8.2% ~ 63.1%, and 22.7% ~ 73.2%, respectively, with a comparable or better accuracy of -0.07% ~ +0.57%, -0.04% ~ +0.32%, and -0.08% ~ +0.34%, respectively, compared with SBM. For example, when training ResNet-74 on CIFAR-100, the Auto-PFQ with a precision schedule of FW(3,4,6,8)/BW(6,8,12,16) achieves 73.2% computational savings over SBM with FW-8/BW-16, with a better (+0.28%) accuracy. Second, progressive quantization along the training trajectory, i.e., PFQ, is in general effective towards efficient DNN training regardless of the precision schedule designs. For example, in the experiments corresponding to Fig. 1, all the

three PFQ variants can reduce the training cost, while not hurting, or even improving, the accuracy, **under different precision schedule schemes with both heuristic and principled designs**.

## 2 FracTrain on larger and deeper models

To verify the scalability of FracTrain on larger and deeper models, we further apply FracTrain to ResNet-110/ResNet-164 on CIFAR-10/CIFAR-100 and find that again FracTrain consistently outperforms the FW8/BW8 baseline (SBM [1]) with 38.69% ~ 67.3% computational savings under a slightly higher accuracy (+0.05% ~ +0.25%).

Table 1: Comparing FracTrain with SBM [1] on ResNet-110/164 on CIFAR-10/100.

| Method | ResNet-110 | | ResNet-164 | |
|---|---|---|---|---|
| | CIFAR-10 | CIFAR-100 | CIFAR-10 | CIFAR-100 |
| FW8/BW8 | 93.38 | 72.11 | 93.72 | 74.55 |
| FracTrain | 93.51 (↑0.13%) | 72.19 (↑0.08%) | 93.77 (↑0.05%) | 74.8 (↑0.25%) |
| Comp. Saving | 67.3% | 45.17% | 38.69% | 43.6% |

## 3 Visualization of bit allocations in FracTrain

**Settings.** In Fig. 2, we visualize the bit allocations of FracTrain ($cp$-1.5/2/2.5/3) at the 5-th, 45-th, and 85-th epoch across all the blocks (one block shares the same precision option) on ResNet-38/CIFAR-100, under which FracTrain achieves a 0.04% higher accuracy and 61.3% reduction in computational cost over the baseline SBM [1] (see the main content's Table 3). Note that due to the input adaptive property of FracTrain, here we show the precision option with the highest probability to be selected by each block averaging over all the images (a total of 50000) from the training dataset.

**Observations and insights.** First, at the early training stage when FracTrain is specified with a small target $cp$, FracTrain allocates more bits to the shallow blocks of the network with smaller widths (i.e., number of output channels), which seems to balance the lighter computation in those blocks. This observation is consistent with that in the SOTA layer-wise quantization work HAQ [2] under constrained model size. Second, as FracTrain learns to switch to a larger target $cp$ towards the end of the training, more bits will be allocated to the last several blocks for better convergence. We can see that FracTrain automatically learns to balance the task accuracy and training efficiency during training by allocating dynamic bits progressively along the training trajectory and spatially across different blocks.

Figure 2: Bit allocations of FracTrain ($cp$-1.5/2/ 2.5/3) on ResNet-38/CIFAR-100 at different training epochs: (a) 5-th, (b) 45-th and (c) 85-th epoch.

## 4 Detailed training settings on CIFAR-10/100, ImageNet, and WikiText-103

**Model structure and optimizer.** For ResNet-18/34, we follow the model definition in [3]; and for ResNet-38/74, we follow the model definition in [4]. For MobileNetV2 on CIFAR-10/100 and the Transformer-base model on WikiText-103, we follow the ones in [5] and [6], respectively. For the

experiments on all the datasets, we adopt an SGD optimizer with a momentum of 0.9 and a weight decay factor of 1e-4, following [7].

**Training on CIFAR-10/100.** We adopt a batch size of 128, and a learning rate (LR) is initially set to 0.1 and then decayed by 10 at both the 80-th and 120-th epochs among the total 160 epochs, as in [8].

**Training on ImageNet.** We adopt a batch size of 256, and the LR is initially set to 0.1 and then decayed by 10 every 30 epochs among the total 90 epochs, following [8].

**Training on WikiText-103.** We train the basic transformer [9] on WikiText-103 consisting of 100M tokens and a vocabulary of around 260K. We use a dropout rate of 0.1, and the Adam optimizer [10] with $\beta_1 = 0.9$, $\beta_2 = 0.98$ and $\epsilon = 10^{-9}$. Each training batch contains a set of 1024 tokens with a sequence length of 256. We train the model for a total of 50,000 steps, following [11].

## 5 Settings of FracTrain on adaptation & fine-tuning scenarios

To evaluate the potential capability of FracTrain for on-device learning, we consider training settings of adaptation and fine-tuning, defined as:

- **Adaptation.** We split the CIFAR-100 training dataset into two non-overlapping subsets, each contains 50 non-overlapping classes, and first pre-train the model on one subset using full precision. Then starting from the pre-trained model, we retrain it on the other subset to see how efficiently they can adapt to the new task. The same splitting is applied to the test set for accuracy validation.

- **Fine-tuning.** We split the CIFAR-100 training dataset into two non-overlapping subsets, each contains all the classes. Similar with adaptation, we first pre-train the model on the first subset using full precision, and then retrain it from the pre-trained model on the other subset, expecting to see the continuous growth in performance. We use the same test set for accuracy validation.

## 6 More details for the experiments of the main content's Fig. 5

In Fig. 5, each experiment result corresponds to one $cp$ setting which ranges from 1% to 6% for experiments with ResNet-38/74 and 3% to 6% for experiments with MobileNetV2.

For the experiments with ResNet-38/74, DFQ considers seven precision options (including FW-0/BW-0, FW-2/BW-6, FW-3/BW-6, FW-4/BW-6, FW-4/BW-12, FW-6/BW-8, and FW-6/BW-12); and for the experiments with MobileNetV2, DFQ adopts five precision options (including FW-0/BW-0, FW-4/BW-8, FW-6/BW-8, FW-6/BW-10, FW-8/BW-8), where FW-0/BW-0 means skipping the computation of the whole block and reusing the activations from the previous layer/block as SLU [5].

## 7 Determine $cp$ for FracTrain's different stages

Here we explain how to determine $cp_i$ in Algorithm 2, which corresponds to the $cp$ value for FracTrain's different stages for a given overall goal of $cp$ (computation percentage over the full precision models and denoted as $cp_{total}$ hereafter) for the whole training process. Specifically, we adopt a simple and intuitive strategy to derive $cp_i$ from $cp_{total}$:

$$cp_{total} = \frac{1}{M} \sum_{i=0}^{M-1} cp_i \ , \quad where \ cp_{i+1} = cp_i + \Delta_{cp} \tag{1}$$

where $M$ is the total number of stages. In this principled way, we can easily determine the $cp$ value of different stages for achieving the specified $cp_{total}$. In all our experiments, we simply adopt a step size of 0.5 , i.e., $\Delta_{cp} = 0.5$. Once $cp_i$ for the $i$-th stage is specified, $\beta$ in Eq.(2) will adaptively flip its sign to achieve the $cp_i$ constraint.