[Reviews · NeurIPS 2020]

Review 1

Summary and Contributions: The paper proposes two methods of mixed precision training methods (methods which combine different numerical precision during training). The first is Progressive Fractional Quantization (PFQ), which uses low precision early in training and increases the precision as training goes on. This reduces quantization noise as over the course of training, similar to how SGD lowers the learning rate over training to reduce the stochastic step noise. Specifically, the authors train the model at a certain precision until the loss has roughly converged (as measured by loss difference in the previous 5 epochs), then increase the precision. The second method introduced is Dynamic Fractional Quantization (DFQ), which dynamically adjusts the precision in each layer based on the current input. This is similar to some architecture exploration methods which learn the precision per-layer (e.g. HAQ from Kuan Wang et. al.). However, in this paper the precision at each layer can change on every input. Each layer is accompanied by a lightweight RNN which predicts the precision to be used. The loss function has an additional term for the total compute cost of the network, which is trained to go towards a user-selected target.

Strengths: The novelty and evaluation results are both compelling. The ideas and data deserve to be seen by others in the field. Both PFQ and DFQ are straightforward and practical quantization techniques which has a chance of being implemented in real systems (especially PFQ). This is a big strength: many quantization methods are too complex to work in hardware. The paper is well written and easy to understand. Anyone reasonably familiar with DNN quantization can follow the paper, even the details. The use of a cycle-accurate simulator to get energy numbers is very appreciated, especially at NeurIPS.

Weaknesses: The biggest limitation in the work is in the evaluation, which largely focuses on CIFAR ResNets with some small ImageNet ResNets (18 and 34), and Transformer-Base. All of these are considered relatively small models in their respective tasks, and far from SOTA. This is a paper on efficient training techniques, which discusses in its impact section the need to reduce ecological footprint of training. The worst offenders are industrial text models such as BERT or GPT. that are being trained by industry labs (OpenAI, NVIDIA, etc). NVIDIA's Megatron paper discusses how even fp16 must be carefully applied in these models (it requires at the very least a dynamic loss scale). I don't have the confidence that techniques in this paper can work on larger models being used in practice. I think it's unreasonable to ask the authors to train something like GPT, but ablation data on on deeper and larger models would be very helpful. EDIT: The authors have addressed both points in their rebuttal. Regarding evaluation on larger models, they chose to add ResNet-110/164 for CIFAR-10. I'd really like to see ImageNet, but this is still a good addition

Correctness: The biggest methodology issue is that the work compares the effective MACs of a mixed-precision (MP) system vs. an integer-only system. Hardware must pay a penalty in area and energy efficiency to support multiple bitwidths, and comparing MACs here is not fair. The authors seem to be getting energy numbers for the integer baselines using a MP hardware system (BitFusion), which is overkill. Ideally, we should compare MP techniques on BitFusion to integer techniques on integer hardware. Without acknowledging this issue in the text, the MAC comparison and energy numbers are extremely misleading. For DFQ, the evaluation compares baselines trained on FW8/BW8 vs DFQ which can select up to BW10 or BW12. This adds to the above point and seems unfair. It's like comparing int8-training to an int8-training + int12 finetuning. These details are on lines 87-89 of the supplementary and not in the main paper, which I think is misleading. EDIT: The authors have performed more experiments to address both my points in the rebuttal. For the integer-only comparison, they simulated a baseline on Eyeriss (a well-known integer-compute architecture) while using the same memory size as BitFusion. The results show PFQ still achieves large energy and latency savings. For DFQ's bit allocation, the authors restricted DFQ to at most 8-bits, and showed that DFQ no longer improves accuracy, but still reduces computation. I'm happy with how these comments are handled and I've bumped my score to a 7. But please move the description of DFQ selecting up to 12 bits to the main paper (caption of Figure 4). Readers may be misled to think DFQ will boost accuracy without additional bits.

Clarity: Yes, I thought DFQ could be explained in simpler terms, but overall the paper is well written.

Relation to Prior Work: Yes. a more explicit discussion of how the paper relates works like Hardware-Aware Automated Quantization (Kuan Wang et. al.) and Incremental Network Quantization (Aojun Zhou et. al.) would be useful. Still, I had no issues seeing the novelty.

Reproducibility: Yes

Additional Feedback: Typo: Line 149: qunatized The space dedicated to CIFAR-10/100 data seems unnecessary. CIFAR ResNets are fairly easy to quantize and I don't really consider it a good benchmark for quantization. Maybe get rid of ResNet-34 for CIFAR-10 and free up some space.


Review 2

Summary and Contributions: The authors resolved my major concerns on 1) hyper-parameter sensitivity and 2) MACs calcualtion, so I raised my overall score to 6. However, I retain the argument that dynamic hardware (especially PFQ) for training provides benifits only to a limited set of ML accelerators and can hardly prove its convergence theoretically and that is why I cannot give a 7. ------------ The paper presents an efficient training method named FracTrain. FracTrain focuses on reducing the precision alone the training trajectory and also dynamically switches to varying precisions based on input values. The evaluation of the paper shows dynamic precision control of DNN training can offer significant computation and latency savings on popular vision and language tasks.

Strengths: 1. The paper focuses on low precision on-device learning, which is an important topic for future edge ML hardware deployments. 2. The paper has a rich collection of experiments and empirically demonstrated the effectiveness of the proposed method on a wide range of datasets and models. 3. The hypothesis of using low-precision at the start of training is similar to using large learning rate is interesting.

Weaknesses: 1. The PFQ algorithm introduced many hyperparameters, and I am curious how the authors chose the parameters \epsilon and \alpha. The authors simply claimed these parameters are determined from the four-stage manual PFQ from Figure 1, and then claim that FracTrain is insensitive to hyperparameters. First, the precision choices of the four stage PFQ in Figure 1 is already arbitrary. Second, I do not think the empirical results can support the claim that FracTrain is insensitive to hyperparameters. I would encourage the authors to have an ablation study of \epsilon and \alpha. I do understand an ablation study of various precision combinations is shown in the appendix, but this might not provide enough insights for users of FracTrain that simply want to know what is the best hyperparameter combination to use. 2. I found the MACs and Energy results reported in the paper needs further explanation. For instance, in Table2, it seems to me MACs cannot be a useful measurement since SBM and FracTrain might use different precisions. Even if the MACs numbers are the same, low-precision operations will surely be more energy efficient. A more useful measurement metric might be bitwise operations. In terms of the Energy reported in this paper, the authors claim it is calculated from an RTL design. However, BitFusion is an inference accelerator, what modifications have you done to the BitFusion RTL to support this training energy estimation? What is the reuse pattern for gradients/activations? 3. Dynamic precision control during training might only show meaningful performance gains on bit-serial accelerators. However, most existing ML accelerators tend to use bit-parallel fixed-point numbers, this might restrict the implications of the proposed methodology. 4. I think this paper has missed a number of citations in the recent advances of dynamic inference methods. Dynamic channel control [1,2] and dynamic precisions [3] have recently been widely explored and these citations are not seen in this paper. [1] Gao, Xitong, et al. "Dynamic channel pruning: Feature boosting and suppression." ICLR 2018. [2] Hua, Weizhe, et al. "Channel gating neural networks." Advances in Neural Information Processing Systems. 2019. [3] Song, Zhuoran, et al. "DRQ: Dynamic Region-based Quantization for Deep Neural Network Acceleration." ISCA 2020

Correctness: Please refer to the weaknesses 1 and 2.

Clarity: I think the paper is well written and easy to follow. There is a decent amount of graphical illustration to help readers to understand, and the experiment results are well displayed.

Relation to Prior Work: This work introduces two axes of dynamic precision control during the training process. Although some of the dynamic control methods (like DFQ) were usded in inference before, I think it is the first time that they were applied to training.

Reproducibility: Yes

Additional Feedback:


Review 3

Summary and Contributions: The authors make the observation that the required precision of weights, activations and gradients for making learning progress is lower at the beginning and reaches full precision towards the end. Not training with full precision until the end of training can therefore save computation when integrating across the whole training run. Subsequently the authors present DFQ, an algorithm that adapts precision to the input by introduction of a gating mechanism. Finally, the authors combine both. ======= I have read the rebuttal and discussed with the other reviewers and AC

Strengths: The strength of this work lies in its empirical evaluation. Given the immense computing power required to train large models, this work is significant in trying to tackle this issue. Once supporting hardware becomes available, techniques like these will accelerate training.

Weaknesses: While highly significant, the hardware to make use of the proposed approaches doesn't exist yet. Apart from prototypes (eg. [32]), current deep learning accelerators cannot harness the theoretical energy savings from (changing) heterogeneous bit-widths.

Correctness: The authors use "the cost metric defined in [30]", however I couldn't find a reference in [30]. Maybe there is a citation issue? I would like to understand how the bit-width of operations influences the cost metric if MACs are counted. According to my understanding, MACs do not by definition take bit-width into account. I would like to understand the train/eval/test split used for datasets. It is not explicitly mentioned and the "Training trajectories" section in 4.4 suggests the authors are plotting test-split numbers.

Clarity: yes

Relation to Prior Work: Yes

Reproducibility: Yes

Additional Feedback: I would like the authors to specify the MAC computation as dependent on bit-width and specify the train/eval/test split. Is the RNN predicting a bit-width per data-point or per mini-batch? If it is per data-point, I would ask the authors to discuss the consequence for hardware accelerators that make use of the parallelizability of NN training across data-points. How is this supposed to be implemented in practice? If these points are addressed I will raise my score.

[Author Response · NeurIPS 2020]

We thank all reviewers for their insightful comments, and have addressed them below.

[R1] **FracTrain on larger and deeper models:** Thanks for the ad-
vice which helps to strengthen our evaluation. Given the limited
time, we apply FracTrain to ResNet-110/ResNet-164 on CIFAR-
10/CIFAR-100 and find that again FracTrain consistently outperforms

| Method | ResNet-110 | | ResNet-164 | |
|---|---|---|---|---|
| | CIFAR-10 | CIFAR-100 | CIFAR-10 | CIFAR-100 |
| FW8/BW8 | 93.38 | 72.11 | 93.72 | 74.55 |
| FracTrain | 93.51 | 72.19 | 93.77 | 74.8 |
| Comp. Saving | 67.3% | 45.17% | 38.69% | 43.6% |

the FW8/BW8 baseline with 38.69%-67.3% computational savings under a lightly higher accuracy (+0.05%- +0.25%).

[R1] **MP on BitFusion and Integer on integer hardware:** We will clarify the evaluation metrics in the final version.
We used Bit-Fusion for both the MP and integer-only systems to keep the hardware parameters (e.g., dataflows) the
same for a fair comparison. We have conducted experiments to address your suggestion by comparing FracTrain on
Bit-Fusion and FW8/BW8 on an integer-only hardware Eyeriss [Y. Chen, ISCA'16] based on the simulator in Tetris [M.
Gao, ASPLOS'17]: for ResNet-38/74 on CIFAR-100 (accuracy in Table 3), FracTrain on Bit-Fusion still outperforms
FW8/BW8 on Eyeriss with +65.8%/+69.8% energy savings and +72.6%/+68.2% latency savings, when both adopting
the same unit energy and memory size as in Bit-Fusion for a fair comparison.

[R1] **Bit choices of DFQ:** It is in fact DFQ's advantage to allow adaptive allocation of higher precision to important
layers/inputs and lower precision to unimportant ones, and thus enable a larger range of precision choices over static
quantization, given the same computational cost. Furthermore, we follow your suggestion, and limit BW in DFQ no
more than 8 bits: compared with FW8/BW8 on (1) ResNet-74@CIFAR-10 (93.04%) and (2) ResNet-74@CIFAR-100
(71.01%), DFQ still achieves slightly higher accuracy (+93.11%/71.11%) with +37.3%/+43.7% computational savings.

[R3] **Sensitivity to hyper-params in PFQ:** Figure 2 in the ap-
pendix shows PFQ's insensitivity to its precision schedule hyper-
params under three different precision schedule strategies. Further-
more, we perform your suggested ablation study to evaluate PFQ-
FW(3,4,6,8)/BW(6,6,8,8) on ResNet-38@CIFAR-100 under various
$\epsilon$ (different shapes) and $\alpha$ (different colors): We can see that a good
accuracy-efficiency trade-off can be found in a large range of settings
compared with static baselines, showing PFQ's insensitivity to hyper-
params. It is intuitive that (1) $\epsilon$ and $\alpha$ control the accuracy-efficiency
trade-off, and (2) a larger $\epsilon$ and $\alpha$ (i.e., faster precision increase) lead
to higher training cost and higher accuracy.

[R3] **Modifications on Bit-Fusion:** We did not modify the BitFusion RTL. As the backpropagation can be viewed
as two convolution processes (for computing error and gradient, respectively), we estimate energy by executing
the three convolution processes of training sequentially in BitFusion. The reuse patterns optimized by BitFusion is
output-stationary for both gradients/activations.

[R3, R4] **How MACs are calculated:** Inspired by the computation complexity determined by precision in Sec-2.1 of
[30], we calculate the effective MACs of the dot product between a and b using (# of $MACs)*Bit_a/32 * Bit_b/32$,
following [J. Shen, AAAI'20], which is in proportional to bit operations. We will clarify this in the final version.

[R3, R4] **ML accelerators to support FracTrain:** Both dedicated ASIC (e.g., [H. Yoo, ISSCC'19] and [H. Yoo,
JICS'20]) or FPGA accelerators (e.g., EDD [Y. Li, DAC'20]) can help exploit FracTrain's best potential by making use
of its lower average precision to save both data movement and computation costs during training. After the submission,
we have proceeded to implement FracTrain on FPGA to evaluate its real-hardware benefits, following the design in
EDD, which adopts a recursive architecture for mix precision networks (i.e., the same computation unit is reused by
different precisions) and a dynamic logic to perform dynamic schedule. Evaluated ResNet-38/ResNet-74@CIFAR-100
on Xilinx ZC706 (accuracy in Table 3), FracTrain leads to 34.9%/36.6% savings in latency and 30.3%/24.9% savings in
energy compared with FW8/BW8. We will clarify this experiment in the final version.

[R3] **Support by bit-parallel accelerators:** Since the precision granularity in DFQ is block/layer-wise (e.g., a block
with several layers in ResNet will use the same precision), bit-parallel is feasible within each block/layer (see our
answer and FPGA implementation right above this one).

[R4] **Dataset split:** We use standard train/test datasets (50000 vs. 10000) for CIFAR-10/100 WITHOUT any special
split. The training trajectories in Figure 5 visualize the evolution of test accuracy during the whole training process,
where the x-axis captures the total computational cost to reach the current epoch instead of the number of epoch. We
will clarify this point in the final version.

[R4] **Granularity of the RNN controller:** We use per mini-batch for hardware-friendly run-time quantization in both
training and inference. We will clarify this in the final version.

[R1, R3, R4] **Typos and missing references:** Thanks a lot for pointing out! We will ensure that they are addressed and
proofread more carefully before camera-ready.

[Meta-Review · NeurIPS 2020]

All three referees support accept. The authors did a good job clarifying the concerns of the reviewers in the rebuttal period. AC also thinks that the experimental results are solid and the idea is worth to share to the community. A potential weakness is that dynamic precision training on bit-parallel hardware is hard, and AC suggests the authors to provide some comments on challenging hardware design aspects in the final draft. This would make the paper stronger and more useful in practice. Irrespectively, AC recommend acceptance.